# Phosphoserine Functionalized Cements Preserve Metastable Phases, and Reprecipitate Octacalcium Phosphate, Hydroxyapatite, Dicalcium Phosphate, and Amorphous Calcium Phosphate, during Degradation, In Vitro

**DOI:** 10.3390/jfb10040054

**Published:** 2019-11-27

**Authors:** Joseph Lazraq Bystrom, Michael Pujari-Palmer

**Affiliations:** Applied Material Science, Department of Engineering, Uppsala University, 75121 Uppsala, Sweden; j.lazraq@gmail.com

**Keywords:** phosphoserine, phosphoserine modified cement, calcium phosphate, degradation, compressive strength, tissue adhesive, tissue engineering, scaffold, biomaterial, mineralization

## Abstract

Phosphoserine modified cements (PMC) exhibit unique properties, including strong adhesion to tissues and biomaterials. While TTCP-PMCs remodel into bone in vivo, little is known regarding the bioactivity and physiochemical changes that occur during resorption. In the present study, changes in the mechanical strength and composition were evaluated for 28 days, for three formulations of αTCP based PMCs. PMCs were significantly stronger than unmodified cement (38–49 MPa vs. 10 MPa). Inclusion of wollastonite in PMCs appeared to accelerate the conversion to hydroxyapatite, coincident with slight decrease in strength. In non-wollastonite PMCs the initial compressive strength did not change after 28 days in PBS (*p* > 0.99). Dissolution/degradation of PMC was evaluated in acidic (pH 2.7, pH 4.0), and supersaturated fluids (simulated body fluid (SBF)). PMCs exhibited comparable mass loss (<15%) after 14 days, regardless of pH and ionic concentration. Electron microscopy, infrared spectroscopy, and X-ray analysis revealed that significant amounts of brushite, octacalcium phosphate, and hydroxyapatite reprecipitated, following dissolution in acidic conditions (pH 2.7), while amorphous calcium phosphate formed in SBF. In conclusion, PMC surfaces remodel into metastable precursors to hydroxyapatite, in both acidic and neutral environments. By tuning the composition of PMCs, durable strength in fluids, and rapid transformation can be obtained.

## 1. Introduction

Calcium phosphate cements have favorable tissue compatibility, inflammation, healing, and remodeling responses when implanted in hard tissues, due to having a similar surface chemistry as the mineral phase of bone (hydroxyapatite) [1,2,3]. Calcium phosphate cements are effective biomaterial scaffolds, facilitating progenitor and stem cell attachment, and formation of new tissue on the cement surface (osteoconduction) [3]. However, the physical and biological properties of synthetic cements are still inferior to native tissue [4,5]. Cements are randomly-organized networks of entangled crystals, while bone tissue is a hierarchically-organized composite, combining mineral and organic phases [6,7,8], with high compressive strength due to organized, nanoscale calcium phosphate [9], and high tensile strength and toughness due to organized, atomic scale, interactions between the organic phase and mineral phase [8,10]. Many attempts have been made to recreate the biological and mechanical properties of bone tissue using a simple additive approach [11,12,13,14,15]. The most common cement additives are calcium chelators, including organic acids (i.e., citric acid) and inorganic polyphosphates (i.e., pyrophosphate) [12,13]. These chelators improve the mechanical strength by reducing the crystal size, and extending the working/setting time. However, chelators produce a monomeric organic phase that binds to the mineral surface [16,17,18,19,20], rather than a molecularly intermixed composite [6,10] with macroscale interactions (i.e., polymeric or crosslinking) [21,22,23,24]. 

Physiochemical modifications can improve mechanical strength by reducing the macroporosity (i.e., controlling particle size) [25,26,27], or microporosity (i.e., sintering,) [28,29]. Chemically-functionalized cements can produce up to a four-fold increase in compressive and tensile (diametral) strength (70 and 15 MPa, respectively) [1,24,30,31], by incorporating synthetic monomers/polymers (poly vinyl chloride, poly acrylic acid, etc.), which can be subsequently crosslinked to produce macroscale, mechanical reinforcement. These synthetic oligomers can also improve the toughness and strength of cements, at the expense of dissolution and resorption rates [21,22,23,24,31,32,33].

Amino acids are an attractive alternative to synthetic monomers, organic acids, or chemical modification. The working/setting time, cement crystal size, and preferentially oriented crystal growth can be finely controlled with amino acid additives [13,34,35,36,37]. Amino acids can also improve the osteoconductivity (cell attachment, survival and proliferation) of cements [16,38,39]. Interestingly, amino acids can create macroscale disorder in cements, via adsorption to crystal surfaces or direct incorporation into the crystal lattice, thereby potentially increasing the dissolution rate, and release of bioactive ions (i.e., calcium, phosphate) [10,17,40,41]. Synthetic amorphous calcium phosphate rapidly converts to metastable phases (i.e., octacalcium phosphate (OCP)), or directly to hydroxyapatite, after exposure to liquids [42,43,44,45]. As a result, the benefits of amorphous calcium phosphate cements in vivo, on resorbing and mineralizing cells recruited to the cement surface (i.e., the release of bioactive ions) are relatively short lived. 

The amino acid phosphoserine plays a pivotal physiological role in calcium sequestration [46], mineralization and calcium phosphate precipitation [47], and tissue adhesion/cohesion [48]. Acidic non-collagenous proteins involved in mineralization in vertebrates, such as bone sailoprotein (BSP), osteopontin (OPN), and dentin phosphoprotein (DPP), are rich in phosphorylated and carboxyl- containing amino acids [47]. Paradoxically, phosphoserine can both accelerate and arrest precipitation/mineralization. In solution, phosphoserine chelates calcium and binds directly to crystal faces to interrupt crystal growth [37,47]. When phosphoserine is bound, embedded as matrix components (i.e., phosphatidyl-serine [46,49], or within proteins in saliva (i.e., statherin [50,51]) or dairy products (i.e., casein [52,53])), it can prevent mineralization in supersaturated fluids by chelating calcium, stabilizing amorphous calcium phosphate complexes, and direct binding to crystal surfaces. Alternatively, bound phosphoserine can facilitate mineralization by increasing the local concentration of sequestered ions, and ionic bonding [54,55]. Mutation studies have confirmed that phosphorylation is required for these activities, and acts by increasing the binding affinity an order of magnitude [51,54,56,57]. Phosphoserine also participates in the dissolution of calcium salts by directly etching the crystal surface, leading to substitution of surface inorganic phosphates with organic phosphate, and ionic chelation via the formation of calcium phosphoserine salt [56,58,59].

We have recently reported that phosphoserine can create disorder in αTCP cements, producing partially, and completely, amorphous cements that remain stable (do not convert to hydroxyapatite) in liquids [60]. This was the first report of an amorphous cement, which remained stable in liquids. Phosphoserine also markedly improved the physicochemical properties of αTCP cements, increasing the compressive strengths from 5 MPa to 50–100 MPa, when up to 50% of the mass was substituted with amino acid. Phosphoserine-modified cements (PMCs) exhibited both atomic-scale interactions, between the mineral and organic phase (detected by NMR), and macroscale interactions within the organic phases (hierarchical organization, self-assembly, and the formation of a non-covalent organic network within the cement), which may account for the high compressive strength. Cements containing phosphoserine also remodel into new bone rapidly, without a pathological inflammatory or healing response [38,39,61]. In contrast, unmodified αTCP cements transform into hydroxyapatite, which requires years to remodel completely, in vivo. 

Most importantly, PMCs display a completely novel property for cements: strong adhesion to biomaterial and tissue surfaces [60,61]. As an adhesive, PMCs can effectively adhere to, and reconstruct, calcified [62] and soft tissue injuries [63], ex vivo. However, before testing PMCs in vivo, the degradation and dissolution properties must be investigated in vitro, and ex vivo. While a tetracalcium phosphate (TTCP)-based PMC has demonstrated excellent biocompatibility, resorption and remodeling into bone, in vivo [61], there are no published in vivo studies on αTCP-based PMCs. The purpose of the present study was, therefore, to investigate the degradation/ dissolution behavior of PMCs in physiological fluids, ex vivo. The cell-mediated resorption rate, in vitro [64] and in vivo [65], is directly related to phase solubility. Since phosphoserine stabilizes metastable phases (αTCP) in liquid, by preventing or delaying dissolution/reprecipitation of αTCP, it is necessary to determine whether the in vivo dissolution and remodeling rate is also delayed. Changes in mechanical strength, due to hydrolysis or dissolution, were evaluated in three different PMCs, over four weeks, in phosphate buffered saline (PBS). Since PMC can be used as both a cement/scaffold (as a thick bulk) and an adhesive (thin layers, 50–500 um), the effective surface area and volume is expected to vary greatly, depending upon the clinical application. The amount of resorption (mass loss) of PMCs has been compared for very thin (high surface area to volume ratio (SVR)), and thick (high SVR) sample discs. Degradation studies have been carried out in three different fluids that mimic distinct environments within tissues: active sites of resorption in vitro (pH 2.7, 0.2 M HCl) [66]; moderately acidic, buffered conditions similar to the pH at resorption sites in vivo (1 M Tris, pH 4.0) [66,67]; and human plasma (SBF, pH 7.4).

## 2. Results

### 2.1. Wet Compressive Strength of PMC Formulations

#### 2.1.1. Sample Formulations for Compression Testing

PMC formulations (Table 1) were selected based upon prior testing. Briefly, PMCs with 30–50% mole% phosphoserine displayed the highest adhesive (shear) and compressive strength after curing for 24 h [60]. Therefore, the ratio of phosphoserine to calcium salt was fixed at 31%:69% (mole%), and the type of calcium phosphate was varied. In formulation 1 only αTCP was used as the calcium phosphate phase. In formulation 2, 1.2% mole% of αTCP was substituted with Portland cement because we have previously observed that minor amounts of Portland cement improve the adhesive strength [68,69,70]. As an internal control, formulation 3 contained only Portland cement and CS1, in equal proportions as in formulation 2. An additional control group, containing only αTCP cement (group Control), was included, only for diffraction studies, to compare the phase changes in PBS. In formulation 4, 1.2% mole% of αTCP was substituted with calcium metasilicate (CS1), also known as wollastonite, because PMCs containing CS1 displayed excellent setting, handling and adhesive properties on soft tissues [63].

#### 2.1.2. Compressive Strength

The initial compressive strength of PMC after seven days was 47.69 ± 4.69, 45.95 ± 6.36, and 49.21 ± 8.78 MPa, for groups 1, 2, and 4, respectively (Figure 1A, p > 0.99 for all comparisons). Control cement (group 3) required at least seven days before achieving sufficient cohesive strength to allow for polishing and testing, therefore no values were obtained at day 1. When comparing between groups, after 28 days in PBS, PMCs containing calcium metasilicate were slightly weaker than PMC without silicate (group 4 vs. group 1, 38.6 vs. 49.2 MPa, *p* = 0.23). When comparing within groups, the compressive strength decreased by 15%, between 1 and 28 days (38.6 vs. 49.5, *p* = 0.11, indicated by *) for group 4, while remaining unchanged for group 1 (47.7 vs. 49.0, *p* = 1.00) and group 2 (48.2 vs. 44.5, *p* = 0.93). All PMCs were three- to four-fold stronger than control cement, which contained the same relative proportions of αTCP to Portland cement (*p* < 0.01 for all group comparisons to group 3, for each time point, indicated by ††). The force-displacement curves (Figure 1B) were comparable between all PMC groups. Note the displacement values for each curve have been scaled (shifted by approximately 0.1 mm for group 2 and 4), for clarity.

#### 2.1.3. X-Ray Diffraction (XRD)

It has been reported that PMCs containing greater than 16% mole% phosphoserine do not convert to hydroxyapatite after 14 days in water [60]. We now extend this finding up to 28 days in PBS (Figure 2). Cements containing solely αTCP (group Control) converted (partially) to crystalline hydroxyapatite within seven days (Figure 2A). In contrast, phosphoserine arrested the transformation of αTCP into hydroxyapatite for up to 28 days (group 1); crystalline peaks for hydroxyapatite were not observed. An amorphous phase (poorly ordered calcium phosphate) was present in PMCs, as a broadened peak that increased background counts between 20 and 36 degrees in PMC samples. The amorphous portion could not be quantitated by XRD, and prevented accurate, quantitative Rietveld analysis of the crystalline phases. Crystalline hydroxyapatite peaks were detected in control cement containing Portland cement (group 3, Figure 2B), but not in PMCs containing Portland cement (group 2, Figure 2B). 

Interestingly, PMCs containing calcium metasilicate (Group 4, Figure 2C) partially converted to hydroxyapatite in PBS, after 14–28 days. This phase change corresponds in time, roughly, with the decrease in compressive strength for group 4 samples (Figure 1A). It is possible that the decrease in mechanical strength, in group 4 samples, is due to the phase change detected by XRD. No other metastable crystalline phases, such as octacalcium phosphate (OCP), dicalcium phosphate dihydrate (DCPD), or tetracalcium phosphate (TTCP) were observed. The amorphous peak, which is present from 6 to 15 degrees in all diffractograms, is due to the amorphous plastic sample powder holder. Collectively, the present data confirms that PMCs stabilize metastable precursor phases (crystalline αTCP), for weeks in aqueous fluids. PMCs containing calcium metasilicate (group 4) appear to convert from αTCP to HA faster than PMCs containing Portland cement (group 2), or lacking silicate entirely (group 1).

#### 2.1.4. Scanning Electron Microscopy (SEM)

After compression testing, fragments from each group were imaged with scanning electron microscopy. The surface morphology, and internal architecture, was identical for all PMCs (Figure 3A–E). The external surface of PMCs consisted of an amorphous organic layer (Figure 3A), and the internal architecture contained a nanoporous (~200 nm) organic matrix (Figure 3B,C,E), with nanoscale mineral (10–40 nm), as reported previously [60]. Hydroxyapatite crystals were only found in samples from group 3 (Figure 3D). In order to confirm that the sputter coating process did not influence the observed morphology, a sample from group 1 was imaged without sputter coating (Figure 3F). Identical topography was observed, with nanoscale porosity and mineral clusters.

### 2.2. Dissolution and Reprecipitation on PMC Discs

#### 2.2.1. Dissolution in Varied Surface Area to Volume Ratios (SVR)

Since no covalent bonds are present in PMCs [60], remodeling is expected to proceed via both abiotic and cell mediated dissolution in vivo, rather than hydrolysis (degradation) [71,72]. Dissolution of calcium phosphate cements is a surface-mediated process [26,73] that determines the subsequent a) reprecipitation as the cement converts from αTCP to hydroxyapatite, and b) eventual erosion of cements in vivo. Prior studies [60] indicate that phosphoserine prevented surface mediated dissolution and reprecipitation by forming an inhibitory coating on the surface of αTCP grains, which is in agreement with the XRD results of the present study (Figure 2). Samples, with a range (Table 2) of surface area to volume ratios (SVR), were fabricated from a single PMCs formulation, containing αTCP and Portland cement (group 2, Table 1), to compare how the sample dimensions affected dissolution, and whether reprecipitation differs between the surface and the bulk.

The average mass loss for each group, after soaking in the respective fluid for 14 days, is shown in Figure 4A. Since calcium phosphate dissolution is pH dependent, greater mass loss was expected in low pH, dilute solutions [3]. Surprisingly, samples incubated in HCl and SBF produced comparable amount of mass loss (Figure 4A, *p* > 0.80 for HCl vs. SBF comparison, for all SVRs) to SBF. PMCs incubated in 1M TRIS experienced the greatest average mass loss, only for the high SVR group (Figure 4B, *p* = 0.01, 0.02 for 25.72 vs. 16.5 and 9.06 SCR groups, indicated by †). A two-fold acceleration of dissolution, in Tris, has been reported for beta-tricalcium phosphate [71], but the effect of TRIS on αTCP dissolution has not been previously investigated.

Correlation scatterplots of the entire range of samples showed a poor correlation between the SVR and dissolution rate (0 < R^2^ < 0.5) for any samples with an SVR below 20 cm^2^ cm^−3^ (Figure 4C). A similar finding has been reported for hydroxyapatite, where the dissolution rate was not affected by the effective surface area [71]. Samples with higher SVR (>16 cm^2^ cm^−3^) exhibited a stronger correlation between SVR and dissolution rate (Figure 4D, R^2^ = 0.81963 Tris, R^2^ = 0.55724 SBF, R^2^ = 0.65511 HCl). This result suggests that as the thickness of PMC decreases (below 0.5 mm), and the SVR increases above 25 cm^2^ cm^−3^, the mass loss for a given incubation period is likely to accelerate.

Collectively, these results suggest that when PMC is applied as an adhesive in thick layers (>500 um), or implanted as a bulk scaffold/implant (i.e., SVR less than 20 cm^2^ cm^−3^, i.e., as a cement), abiotic dissolution is low enough (8–14% after 14 days) that the compressive and adhesive strength will be maintained for weeks in physiological fluids. Conversely, when PMC is applied as a thin layer adhesive at tissue interfaces, the dissolution rate, and deterioration rate of mechanical (adhesive) strength, may be tuned by varying the thickness and SVR.

#### 2.2.2. X-Ray Diffraction and Infrared Spectroscopy (FTIR-ATR) Analysis

The surface and bulk crystalline phases, present on PMC discs (group 2) from the lowest SVR group soaked for 14 days, are shown in Figure 5A. PMC from the 0.2 M HCl (pH 2.7) group displayed the greatest relative amount of hydroxyapatite formation, indicated by comparing the relative intensity of the 2 1 1 (31.8 degrees) HA peak, to 0 3 4 peak (30.7 degrees) of precursor αTCP. Unfortunately, the presence of an amorphous phase (20–36 degrees), and lack of crystalline reference for phosphoserine, precluded quantitative phase analysis by Rietveld refinement. Therefore, all subsequent comparisons regarding relative quantity refer to comparison of the primary crystalline peak intensity for each phase: hydroxyapatite (2 1 1, 31.8 degrees); octacalcium phosphate (1 0 0 peak, 4.6 degrees); dicalcium phosphate dihydrate (0 2 0 peak, 11.7 degrees),, in comparison to αTCP (0 3 4 peak, 30.7 degrees). Other acidic, metastable, phases were also found in the HCl group, including DCPD, and OCP (Figure 5A). The brushite peak, at 11.6 degrees, displayed significantly greater relative intensity than αTCP (30.7 degrees). Peaks corresponding to calcium phosphoserine (13.7–13.9 degrees) and crystalline phosphoserine (17.37, 19.5, and 26.45 degrees, possibly occurring via recrystallization) were also observed. Interestingly, while high intensity DCPD peaks were present, the highest intensity peaks observed in TRIS samples corresponded to calcium phosphoserine, and no OCP peaks were observed. Hydroxyapatite peaks were not obvious in diffractograms of the surface of PMCs soaked in 1 M TRIS (pH 4.0), though a peak at 33.2, in the region of HA, did not match any known phases. Based upon the relative peak intensities, the surface of TRIS samples contained primarily calcium phosphoserine salt, and a significant amount of DCPD, while HCl samples contained primarily OCP, DCPD, and hydroxyapatite. Formation of hydroxyapatite is thermodynamically favorable near neutral pH. Therefore, it is likely that the formation of hydroxyapatite under acidic conditions (pH 2.7) results from conversion of acidic metastable phases OCP and DCPD [74,75]. Since these phases form readily in acidic conditions and are metastable, hydroxyapatite is expected to form conversion from OCP and DCPD, during the ongoing dissolution/reprecipitation reaction [75,76].

XRD of PMCs soaked in SBF did not contain crystalline HA peaks. A crystalline calcium phosphoserine peak was also observed in SBF samples, though with low relative intensity. No other crystalline phases were observed, suggesting that any other reprecipitated phases must be amorphous or poorly ordered and, therefore, undetected by diffraction analysis. 

Collectively, the results of surface XRD suggest that dissolution of the surface of PMC, in acidic conditions, results in formation of significant amounts of HA [76]. HA reprecipitation occurs on concert with formation of metastable phases DCPD and OCP. Conditions that mimic human plasma (SBF, pH 7.4), did not readily produce HA. The presence of TRIS, which accelerated dissolution up to 2-fold in acidic conditions, appeared to favor formation of calcium phosphoserine, and acidic phase DCPD. Though the rate of PMC mass loss was comparable in neutral and acidic conditions (Figure 4), crystalline reprecipitation phases were only found in significant quantity in acidic conditions. 

Powder XRD of the bulk (Figure 5B) contained peaks for DCPD, hydroxyapatite, and crystalline calcium phosphoserine (12.2–12.4 degrees), under acidic conditions (TRIS and HCl). In contrast to surface XRD, OCP was not detected and the relative intensity of DCPD and HA peaks were much lower than αTCP. In Figure 5B the broad amorphous background, from 4 to 15 degrees is due to the amorphous plastic sample holders. In neutral pH (SBF), the bulk of PMC displayed much lower relative peak intensities for DCPD and HA, compared to TRIS and HCl. Unless indicated otherwise, other crystalline calcium phosphate phases, such as OCP, MCPM, TTCP, or beta-TCP, were not detected.

Collectively, bulk powder XRD analysis indicated that acidic conditions, like those found at sites of active resorption in mineralized tissue, stimulate dissolution and reprecipitation, even in the absence of cells and biomolecules. While the surface of PMC is clearly dynamic, the bulk also dissolves and reprecipitates at a significant, albeit slower, rate. The resorption and remodeling rates are critical properties for hard tissue adhesives. In the body, where numerous other resorption processes occur (i.e., enzymatic and cell mediated degradation), dissolution, reprecipitation and remodeling are expected to proceed at a faster rate than described in the present work. The major phase comprising the PMC bulk remains αTCP, even after 14 days in aqueous conditions that stimulate dissolution and reprecipitation. In vivo, PMCs could act as a stable reservoir as they degrade; providing a steady source of metastable, amorphous calcium phosphate that resorbing, mineralizing, and progenitor cells continually renew as the implant surface is resorbed. The present data indicate that PMCs may combine the benefits of amorphous cements (transformation into metastable phases; source of bioactive ion release) with the stability of crystalline cements.

Interestingly, FTIR peaks corresponding to the carbon- (2935, 2861 cm^−1^), and amino- (3309 cm^−1^), hydrogen bond stretches, and carbonyl (1645 cm^−1^) groups in phosphoserine were only obvious in TRIS spectra (Figure 5C) [77]. Only surface precipitants from TRIS samples, therefore, contained detectable levels of organics (phosphoserine). The very broad peaks at 550–600 and 1020–1150 cm^−1^, in the SBF group (Figure 5D), confirm the presence of amorphous calcium phosphate [78,79]. The sharp peak at 1025 cm^−1^ and doublet at 475 cm^−1^ correspond to crystalline hydroxyapatite, which is present in HCl spectra, and barely discernable within the amorphous peak of SBF samples at 1020–1150 cm^−1^ [78,80]. The broadened peaks at 534 and 570 cm^−1^, present in TRIS and HCl spectra, correspond to DCPD, ν4 PO_4_^3−^, stretching. DCPD peaks, at 630, 800, and 890 cm^−1^, were also visible in TRIS and HCl samples. In HCl samples the peak at 570 was higher absorbance than the 534 cm^−1^ peak. This was due to an overlapping OCP peak 560 cm^−1^, which is confirmed by the presence of clear OCP peaks at 604, 1010; and 1050–1150 cm^−1^ [78,81]. 

#### 2.2.3. Scanning Electron Microscopy

After 14 days in acidic conditions (0.02 M HCl, pH 2.7), cuboid structures (Figure 6A2), and flat, plate like crystals (Figure 6A3) resembling OCP and HA [78], were found on the surface of PMC (Figure 6A1). Phase and crystallinity analysis of the cuboid structure is not possible with SEM or EDS, therefore the term “cuboid morphology”, or “cuboid” is used, hereafter, to describe these structures. Interestingly, the cuboids closely resemble the morphology of phosphoserine that has recrystallized in PMCs containing excessive (>50% mole%) phosphoserine [60]. The presence of TRIS favored the formation of cuboids (Figure 6B1), as these were the only feature visible on the surface of PMC (Figure 6B2). Since the predominant surface phase detected by XRD was calcium phosphoserine and DCPD, it is likely that the cuboids observed in acidic conditions are a mixture of calcium phosphoserine salt, and DCPD. The fine structure of the cuboids differed between HCl and TRIS samples, with HCl cuboids appearing more amorphous/organic and proteinaceous at the border between cuboids, while TRIS cuboids appeared more crystalline. The PMC surface, after incubating in SBF, contained spherical aggregates (Figure 6C1) of small plate like crystals (Figure 6C3), and very few sparse regions of cuboidal crystals (Figure 6C2). Cuboids were only found in regions where the mineralized coating was thinnest.

Collectively, the SEM results suggest that, as calcium and phosphate from the mineral phase of PMC dissolves, cuboids, plates and amorphous regions are present on the remodeling PMC surface. The cuboid morphology appeared, exclusive of other morphologies on TRIS sample surfaces, where calcium phosphoserine and DCPD peaks were detected by surface XRD (Figure 5A). Since cuboids were not observed in PMCs prior to soaking in fluids, cuboids must form during reprecipitation, rather than arising during the curing/setting reaction and, subsequently, being revealed by surface dissolution. While the cuboid morphology appeared similar to reprecipitated crystalline calcium phosphoserine, it is also possible that the plate-like cuboids contain DCPD. The small, plate-like aggregates that covered the entire surface of SBF samples did not correspond to any crystalline peaks in XRD analysis, though they do match amorphous calcium phosphate detected by FTIR. In SBF, PMCs appear to precipitate a poorly ordered, or amorphous, surface coating of calcium phosphate (calcium to phosphate ratio ~1.5), which has been reported to precipitate on other osteoconductive material surfaces exposed to SBF [82,83,84].

#### 2.2.4. Energy Dispersive X-Ray Spectroscopy

The morphologies observed by SEM were further characterized by EDS mapping and spectral (spot) analysis (Figure 7). A “true colored”, elemental map is overlayed on a representative SEM image, in Figure 7A (HCl), 7B (Tris), and 7C (SBF). An area was chosen (Figure 7A) that included distinct regions of plate-like crystals (region 1), cuboids (region 2), and what appears to be either the undissolved remnants of the original amorphous PMC surface, or amorphous regions within the PMC bulk that were revealed by dissolution (region 3). It should be noted that only cuboids and crystals were prevalent, and amorphous regions (region 3) were rarely found. TRIS samples contained only cuboid crystals (Figure 7B, region 4). In SBF samples (Figure 7C) cuboids (region 6) were often subsumed by the plate-like coating (region 5), therefore analysis was conducted on two distinct regions, where no intermixing of these structures occurred. Small sections within the plate-like coating appeared amorphous, possibly reflecting phosphoserine reprecipitation/coating on the surface, or redeposition of carbon during imaging (an artifact). Measurement error was minimized by deconvoluting lighter elements (carbon and oxygen), as well as the sputter coating elements palladium and gold, from the raw spectra obtained from spot analysis (Figure 7D, regions 1–3; Figure 7E, region 4; Figure 7F, regions 5–7). Sodium, magnesium, and zinc were detected in SBF samples, though the amount was below 2% (relative at%), therefore these elements were also deconvoluted from the analysis. 

The amorphous region in HCl (Figure 7A, region 3, Figure 7D, red dotted line) was rich in carbon, nitrogen (amine group in phosphoserine), phosphorus and oxygen, compared to plates (region 1). This result suggests that the amorphous region is a thick, homogenous layer of organics (phosphoserine), and is calcium deficient. In contrast, the amorphous topography observed in the SBF sample (Figure 7C, region 7, Figure 7F, red dotted line), contains identical elemental stoichiometry to small plates, suggesting that SBF amorphous regions are superficial, thin redepositions of organics (phosphoserine), on a thicker underlying mineral layer.

The relative intensity of the carbon/nitrogen peak, compared to oxygen and phosphorus, indicates whether the phosphorous in each sample is organic (greater carbon to oxygen/phosphorus intensity) or inorganic (lower carbon to oxygen/phosphorus intensity). The amorphous region 3 (Figure 7A) contained a high intensity carbon peak, compared to oxygen/phosphorus, suggesting that phosphate is largely organic. The amorphous region spectra also differed from cuboids. Cuboids, in all three samples (regions 2, 4, and 6; Figure 7A–C), contained much higher intensity peaks for phosphorus and oxygen than carbon, suggesting that cuboids contained significant amounts of inorganic phosphate. The amount of nitrogen (relative to calcium and phosphate, Figure 7G–I) in cuboids varied, ranging from 43.05% (SBF) to 35.11% (HCl) to 28.45% (Tris), though the differences were not statistically significant (p > 0.23 for all comparisons). Therefore, it is likely that the relative amount of organic phosphate, and inorganic calcium and phosphate, varies in cuboids, despite exhibiting similar morphology (SEM).

When the EDS spectra from two types of plates (region 1 and 5) were compared, the larger plates (region 1) contained more carbon and nitrogen than small plates (region 5, Figure 7G,I). While large plates contained more organic phosphate, the phosphate in both large and small plates appeared to be, predominantly, inorganic phosphate. When comparing plate to cuboid spectra, large plates (region 1) have comparable intensity for organics (carbon, nitrogen) and higher intensity peaks for inorganic phosphate, than cuboids. Small plates (region 5) have lower intensity for organics (carbon, nitrogen) and higher intensity peaks for inorganic phosphate, than cuboids. 

Collectively, the EDS spectra suggest that the cuboids are a mixture of calcium phosphoserine (organic) and a large amount of inorganic phosphate and calcium, possibly (the only other crystalline phase found with similar morphology) DCPD (Figure 7G–I, regions 2, 4, 6). Despite variable nitrogen content between all cuboids, the calcium to phosphate ratio was identical, at 0.99 ± 0.01, 1.05 ± 0.06, and 1.03 ± 0.02 for regions 2 (HCl), 4 (Tris), and 6 (SBF), respectively (p > 0.491 for all comparisons). The calcium to phosphate ratio of DCPD and calcium phosphoserine are both 1.00, therefore cuboids could be a mixture of these two phases, in agreement with the surface XRD results. Since cuboids with identical calcium to phosphate ratio were found on the surface of all PMC groups, while crystalline DCPD was only detected on TRIS and HCl samples, the inorganic calcium and phosphate content of cuboids may include orthophosphate, in agreement with prior NMR findings and other biological systems that utilize phosphoserine [53,60]. It has also been reported that, as the amino acid content is increased from 0% to 63% mole% in PMCs, the inorganic αTCP phase is converted from crystalline to completely amorphous, while the organic portion always remains amorphous (no crystalline peaks for calcium phosphoserine or crystalline phosphoserine were detected).

The calcium to phosphate ratio in amorphous region 3 was 0.88 ± 02 (Figure 7G), well below other observed crystalline phases DCPD (1.0), OCP (1.33), and calcium-phosphoserine (1.0). This region may represent an area where inorganic calcium and phosphate have dissolved, leaving behind calcium-deficient phosphoserine. The calcium to phosphate ratio of large plates was 1.12 ± 0.04, much lower than would be expected for pure OCP (1.33) or apatite (1.5–1.65). Significant amounts of nitrogen are present in, or on, the large plates (Figure 7G, region 1), but not small plates (Figure 7I, region 5). The presence of organic phosphate would reduce the effective calcium to phosphate ratio in large plates, likely reflecting a mixture of OCP (1.33) and calcium phosphoserine (1.0), or DCPD (1.0), which would produce an effective calcium to phosphate ratio of 1.165 for an equimolar mixture; very close to the observed value of 1.12. EDS analysis suggests that phosphoserine is incorporated directly into/onto these plate-like crystals. This modification could occur via multiple mechanisms, including non-specific binding (surface adsorption), preferential adsorption to crystal faces, or direct incorporation into the lattice [20,35,42].

The calcium phosphate ratio of small plates was 1.41 ± 0.01 (Figure 7I, region 5), significantly higher than for large plates, and very close to the value for tricalcium phosphate (1.5), apatite (1.5–1.65), and amorphous calcium phosphate (1.3–1.5) [43,85]. Based upon the results presented thus far, small plates are consistent with amorphous, or poorly crystalline, calcium phosphate because: (a) quantitation of the EDS spectra shows that no nitrogen was present, only inorganic calcium and phosphate (Figure 7I, region 5); (b) the pervasive quantity of small plates, compared to cuboids (Figure 6B1, SEM) on the surface of SBF samples does not match the low relative peak intensity of HA via surface XRD analysis (Figure 5A, XRD), therefore this is unlikely to be crystalline HA; (c) the morphology (SEM) is significantly different than the precursor (crystalline αTCP or calcium silicate particles) [63], and previously observed final inorganic phases, after curing [60], therefore, this is unlikely to be crystalline αTCP.

## 3. Discussion

Phosphoserine modified cements were the first reported example of partial, or fully, amorphous calcium phosphate cements that remained amorphous in liquids for weeks [60]. While prior investigations of PMCs have focused on the tissue- and biomaterial–adhesion properties [60,61,62,63], the present work evaluated how the delayed dissolution and phase transformation of precursor αTCP phases affected surface dissolution, reprecipitation, and mass loss (degradation). The degradation rate of a tissue adhesive must be slow enough to prevent adhesive failure during the initial stages of healing, when an adhesive must keep injured tissue surfaces appositioned, yet fast enough to allow new tissue to replace the adhesive, otherwise healing will be delayed [86]. Therefore, the dissolution rate was investigated over a range of conditions: in a simple acidic environment that mimics the lowest observed pH at sites of active resorption, in vitro (0.02 M HCl, pH 2.7) [66]; an acidic environment with similar pH to in vivo resorption conditions (1M Tris(hydroxymethyl) aminomethane, Tris-HCl, pH 4.0), and fluid with ionic concentrations that mimic human plasma (simulated body fluids, SBF, pH 7.4) [87,88]. 

The novelty of the present work includes the first report of: (1) PMC dissolution rate, in vitro; (2) surface remodeling and reprecipitation/mineralization in fluids that mimic specific physiological environments (i.e., sites of active resorption, or remineralization); and (3) how the chosen calcium salt affects the physiochemical properties of PMC (i.e., rate of transformation into hydroxyapatite and stabilization of metastable phases like αTCP) in liquids. Calcium metasilicate (wollastonite), but not tricalcium silicate (Portland cement), appeared to accelerate the conversion of αTCP to hydroxyapatite, between days 14 and 28 in PBS. When considering the changes observed in compressive strength and phase composition in PBS, we can conclude that: (a) The delay in dissolution/reprecipitation, in PMCs, preserves the wet mechanical strength in physiological fluids; (b) formulations, or additives, that accelerate dissolution, reprecipitation, or formation of new crystalline phases in PMC, produced only minor reductions in mechanical strength (~10%), which, together with (a), suggests that PMCs are likely to sustain strong adhesive tissue bonding during the initial phases of resorption in vivo; (c) additional studies are needed, to determine how the formulation, additives, and curing conditions, affect the rate of conversion from αTCP to hydroxyapatite, in PMCs.

Dissolution testing in acidic and neutral liquids demonstrated that PMCs may combine the benefits of amorphous and crystalline cements. Phosphoserine stabilized αTCP and prevented conversion into hydroxyapatite. During the process of resorption in vivo, as the surface of PMCs is resorbed, the underlying αTCP phase will be continually exposed, where it can dissolve and reprecipitate, thereby releasing bioactive ions at a faster rate than crystalline phases like beta tricalcium phosphate, or hydroxyapatite. Unlike amorphous calcium phosphates, however, PMCs prevent the transformation of αTCP for weeks, thereby extending the release of ions. 

A second concern addressed in the present study, was whether the delayed dissolution/ reprecipitation of PMCs, where metastable αTCP precursor phase was stabilized in liquids, also negatively impacted bioactivity (formation of hydroxyapatite). Metastable phases, OCP and DCPD, were observed in acidic conditions, as has been seen for other osteoconductive calcium phosphate cements [73]. Surprisingly, HA also formed in acidic conditions. In agreement with this result, Pan et al. has reported that HA may form below pH 4, though likely in a calcium deficient form with lower pH [76]. Bohner et al. has also pointed out that incubation in SBF is not an accurate predictor of in vivo mineralization, and that test conditions can affect the observed rate of hydroxyapatite formation [89]. The dissolution experiment in the present study was designed to identify the optimal conditions for testing (i.e., sample dimensions, and the effect of pH and ionic concentration). Therefore, rather than focusing on bioactivity, instead the focus was evaluating how the surface of PMCs dissolved, and which phases reprecipitated. It should be noted that pH changes in the dissolution fluids were not measured, and ion release was not quantitated over the 14 days. Future work should measure changes in pH, ion release, and try to determine the rate of dissolution (i.e., additional time points), since comprehensive conclusions cannot be drawn based upon a single time point.

While it is clear that PMC dissolves in acids, it is unclear why HCl and SBF samples displayed comparable amounts of mass loss. It is possible that the actual mass loss differs between the two conditions, but that the faster dissolving solution lead to higher ion concentrations and, consequently, faster reprecipitation. Since SBF contained small amounts of Tris, it is also possible that TRIS increased the mass loss in SBF samples. Lastly, dissolution of PMCs may not dependent upon the calcium phosphate phase; instead, phosphoserine dissolution may be the rate-limiting step for PMCs. The present SEM and EDS results suggest phosphoserine is present in the form of plate-like crystals, and as calcium phosphoserine, which form during reprecipitation. 

## 4. Materials and Methods

### 4.1. Materials

All materials were purchased from Sigma-Aldrich (AB Sigma-Aldrich, Stockholm, Sweden), unless otherwise indicated. O-phospho-L-serine, referred to hereafter as phosphoserine, was purchased from Flamma SpA (> 95%, Flamma SpA, Bergamo, Italy). Alpha tricalcium phosphate (αTCP, Ca_3_(PO_4_)_2_) was synthesized as described previously [60]. Portland white cement was purchased from Aalborg Portland (63.9% CaO, 19.5% SiO_2_, Cementir Holding S.p.A, Rome, Italy).

### 4.2. Material Synthesis and Milling

Alpha tricalcium phosphate (αTCP, Ca_3_(PO_4_)_2_) was synthesized by heating (Carbolyte oven CWF1300, AB Ninolab, Stockholm, Sweden) calcium carbonate and monocalcium phosphate anhydrous (MCPA), at a 2:1 molar ratio, on a zirconia setter plate for at 1450 °C, for 12 h [60]. After quenching in air, αTCP powder was dry milled (Reitsch PM400, AB Ninolab, Stockholm, Sweden) in a 500 mL zirconia milling jar, at 300 RPM for 15 min, with 100 g of powder per 100 zirconia milling balls (10 mm diameter). Calcium metasilicate and Portland cement were used, as received. 

### 4.3. Material Chracterization

X-ray diffraction (XRD): The physical properties of αTCP [60,63,68,69,70], calcium metasilicate [63], and Portland cement [68,69,70], have previously been reported. X-ray diffractograms (XRD) of powder samples were obtained on a Bruker D800 Advance (Bruker Daltonics Scandinavia AB, Solna, Sweden), scanned with a step size of 0.03 degrees per step, from 3 to 60 degrees. The phase composition was identified by peak location, in comparison to reference files: PDF# 04-010-4348 alpha tricalcium phosphate (αTCP), #04-008-8714 beta tricalcium phosphate (βTCP), #01-074-0565 hydroxyapatite (HA), #04-011-1625 tetracalcium phosphate (TTCP), #04-013-3344 dicalcium phosphate dihydrate (DCPD), #04-013-3883 octacalcium phosphate (OCP).

Scanning electron microscopy (SEM): Prior to SEM/EDS analysis, samples were sputtered with gold and palladium (10 nm thick, Emitech SC7640, Quorum technologies, Kent, UK), for 40 s at 2 kV. The internal microstructure of samples after compression testing, and on the surface of PMC discs that were soaked in 0.2 M HCl, 1 M Tris, or SBF; were imaged with a Merlin field emission SEM (AB Carl Zeiss, Stockholm, Sweden), with an secondary electron in-lens detector, at an acceleration voltage of 3 keV and a working distance of 5 mm, and 195 pA current. 

Energy dispersive X-ray spectroscopy (EDS) (X-max, AB Carl Zeiss, Stockholm, Sweden): An X-Max 80 mm^2^ silicon drift EDS detector, at 20 keV and a working distance of 8.5 mm, was used for elemental analysis. The data was analyzed with Aztec analysis software (AZTEC 3.3 SP1, Oxford Instruments, High Wycombe, UK).

Fourier transformed infrared spectroscopy (FTIR): The reprecipitated phases, on the surface of discs that underwent dissolution testing, were collected by scraping as described for surface XRD, and analyzed with a Bruker Tensor 27 (Bruker Daltonics Scandinavia AB, Solna, Sweden), with a platinum ATR attachment. Absorbance spectra was collected over the range of 400–4000 cm^−1^.

### 4.4. Sample Preparation

Compression samples were created by hand-mixing powders, as described previously [90], with a defined mole percentage of αTCP to phosphoserine (Pser) (group 1, 31% Pser, 69% αTCP); phosphoserine to αTCP and Portland cement (group 2, 31% Pser, 67.8% αTCP, 1.2% Portland); αTCP to Portland cement (group 3, 98.3% αTCP, 1.7% Portland); or phosphoserine to αTCP and calcium metasilicate (CS1) (group 4, 31% Pser, 67.8% αTCP, 1.2% CS1). Deionized water was added and PMCs were hand mixed with a spatula for 20 s. A liquid-to-powder ratio of 0.2 mL^−1^ g was used for all PMCs (groups 1, 2, and 4), and 0.4 mL^−1^ g for controls (groups 0 and 3). Compression samples, cast in silicon molds (0.6 cm diameter × 1.3 cm length), were prepared from 1.2 g of each PMC formulation, and allowed to cure for 10 min in air before incubating in phosphate buffered saline (PBS, calcium free). 

Degradation samples were made from a single formulation of PMC (group 2, 31% Pser, 67.8% αTCP, 1.2% Portland), and cast in silicon molds of varied diameter and thickness (Table 2). 

### 4.5. Mechanical Testing

Prior to compression testing, sample surfaces were polished to 1200 grit fineness with silicon carbide polishing paper (Struers A/S, Bromma, Denmark). Samples were loaded to failure on a Shimadzu AGS-X mechanical testing machine (Shimadzu Europa Gmbh, Duisburg, Germany), at a crosshead speed of 1 mm per minute, with a 5 kN load cell. The resulting data was analyzed with Trapezium Lite (version 1.0.1) software, provided by the manufacturer (Shimadzu Europa Gmbh, Duisburg, Germany).

### 4.6. Dissolution Testing

Each sample was cured for 24 h in PBS, rinsed in deionized water, and samples were dehydrated for 6 days under vacuum (300 mBar). The dry weight (initial mass) of each sample was recorded after drying for 1, 2, 3, and 6 days. Samples were then placed into the respective dissolution/ mineralization conditions. Preliminary experiments, comparing different drying methods and drying times, confirmed six days was sufficient to remove all non-crystal water from samples.

Disc shaped samples were incubated for 14 days in liquids representing the dissolution environment at sites of active resorption in vitro (0.2 M HCl (pH 2.7) [66,91], or supersaturated blood plasma (SBF, pH 7.4) [88], at a liquid to surface area ratio of 4.424 mL cm^−2^ [87,88,92,93], under static conditions. A third liquid (TRIS buffered water, 1 M Tris-HCl, pH 4.0) was included to represent a buffered acidic environment, in the reported pH range of resorbing osteoclasts in vivo [66,94]. The liquid was refreshed every seven days, and discs were positioned in 50 mL centrifuge tubes to minimize surface contact with the tube, as described in [87]. After 14 days each sample was rinsed in water for 4 h, and dehydrated by vacuum for seven days. The dry weight was recorded, and the mass loss percentage was determined by subtracting the final from the initial dry mass, divided by the original dry mass.

### 4.7. Statistical Analysis

Comparisons between compression (Figure 1A), degradation (Figure 4A), or EDS (Figure 7A) sample groups were evaluated with a one-way ANOVA, with SPSS software (version 22, SPSS Inc., Illinois, USA), using Games-Howell post hoc analysis. In Figure 1A comparisons were made between all compression samples, for each time point (indicated by †); and within each PMC group comparing the strength at each time point to the initial strength on day 1 (indicated by *). In Figure 4A comparisons were made between liquid groups (indicated by *); and within liquid groups, but between SVRs (indicated by †). * and † indicate *p* < 0.05, and ** and †† indicate *p* < 0.01, respectively.

## 5. Conclusions

The rate of dissolution, degradation, resorption, and changes in mechanical strength, are critical properties for hard tissue adhesives. An effective adhesive must maintain the bond strength for days to weeks, to facilitate new tissue formation. Phosphoserine-modified cements retained the benefits of crystalline cements (high compressive strength, and slower dissolution rate), while also presenting amorphous calcium phosphate and metastable αTCP at the cement surface, during dissolution. The addition of minor amounts of calcium metasilicate to PMCs accelerated the formation of hydroxyapatite after 14 days in saline. Finally, PMCs dissolute/degrade in acidic fluids, which mimic the osteoclastic resorption conditions, and in pH neutral fluids with similar ionic concentrations to human plasma. We have shown that the organic layer, which covers the surface of PMC and stabilizes αTCP by preventing dissolution, eventually dissolves in neutral and acidic conditions. As PMC dissolves metastable phases, such as OCP, DCPD, and ACP, appear on the surface and, to a lesser degree, the bulk. It is likely, therefore, that PMCs can be remodeled in vivo, and may stimulate mineralization during the dissolution process via formation of metastable, and amorphous, calcium phosphate.

## 6. Patents

The author M.P.-P. is a co-inventor on the following relevant patents: #WO2019106173, #SE1651271, and # EP3518996/WO2018060289. 

## Figures and Tables

**Figure 1 jfb-10-00054-f001:**
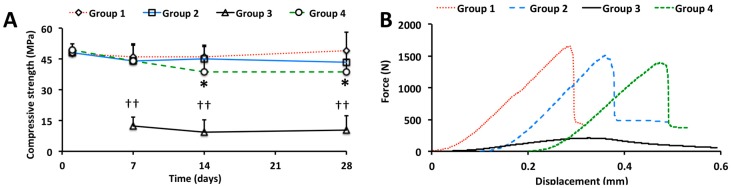
Compressive strength of PMCs. (**A**) The compressive strength of cylinders, composed of PMC containing no silicate (group 1), Portland cement (group 2), or calcium metasilicate (group 4), after curing in PBS. Control cement (group 3), was significantly weaker at each time point (*p* < 0.005 for all comparisons comparing group 3 to all other groups, at all time points), after setting for seven days; (**B**) Representative force/displacement curves showing comparable modulus (slope), and brittle failure for groups 1, 2, and 4. Statistical analysis compared the mean strength between all groups, at each time point (indicated by †); and within each PMC group, the average strength at each time point was compared to the initial strength (indicated by *), using ANOVA with Games–Howell post hoc analysis, with * and † indicating *p*-values below 0.05, and ** and †† indicating a *p*-value below 0.01.

**Figure 2 jfb-10-00054-f002:**
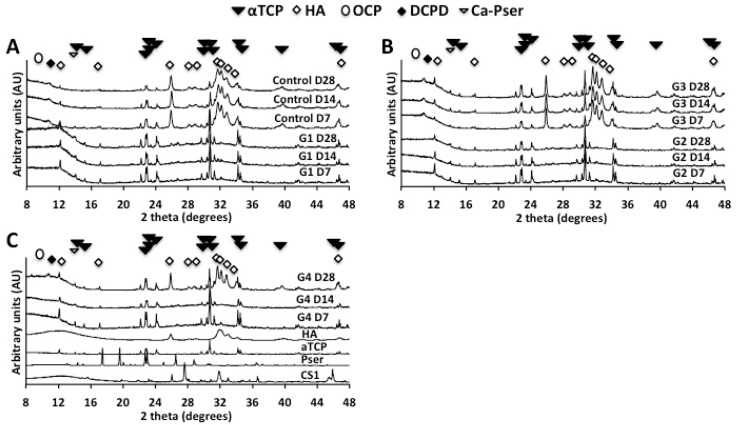
XRD of PMCs after compression testing. (**A**) PMCs without silicate (group 1) did not convert to hydroxyapatite from αTCP after 28 days in PBS, while control cement (group Control) converted to hydroxyapatite within seven days; (**B**) PMC containing 1.2% Portland cement (group 2) did not convert to hydroxyapatite from αTCP after 28 days in PBS, while a second control cement containing 1.7% Portland cement (group 3) converted (partially) to hydroxyapatite with seven days; (**C**) PMCs containing calcium metasilicate (group 4) partially converted to hydroxyapatite, between 14 and 28 days, in PBS. Control cement for group 4 (αTCP and calcium metasilicate, without phosphoserine) did not set, and therefore could not be tested. Abbreviations: HA (hydroxyapatite), αTCP (alpha tricalcium phosphate), Pser (phosphoserine), CS1 (calcium metasilicate, or wollastonite).

**Figure 3 jfb-10-00054-f003:**
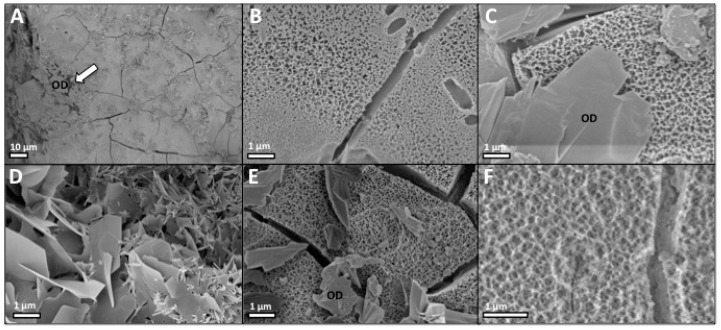
Scanning electron microscopy images of cured PMC cross-sections, after compression testing. (**A**) Fragments from the surface organic layer, seen as organic debris (OD), were observed contaminating the surface in most fields of view (sample from group 1); (**B**,**C**,**E**) Group 1 (**B**), 2 (**C**) and 4 (**E**) exhibited a porous (nanoscale) internal architecture; (**D**) Group 3, which lacked phosphoserine, contained plate and small needle-like crystals (hydroxyapatite); (**F**) The internal architecture of a sample from group 1 was imaged, without sputter coating, to confirm the interconnected, porous architecture observed in the present work, and prior studies, was not affected by the preparation/sputtering process.

**Figure 4 jfb-10-00054-f004:**
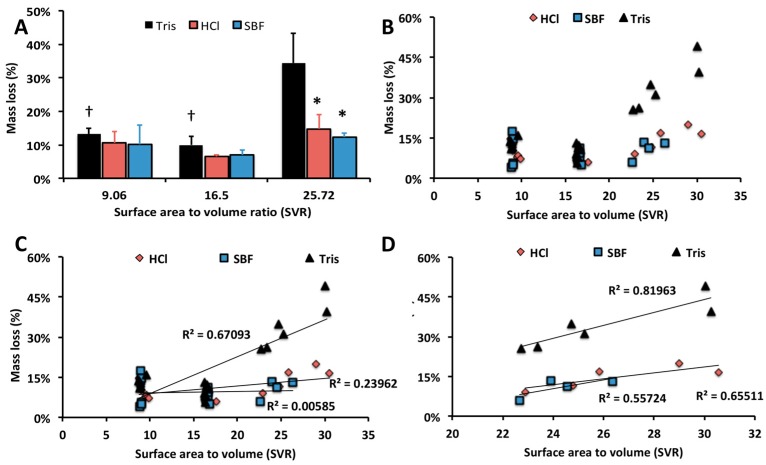
Dissolution and mass loss of PMC discs (group 2) in 0.2 M HCl (pH 2.7), 1 M TRIS (pH 4.0), and SBF (pH 7.4). (**A**) After 14 days all sample groups and SVRs, except TRIS SVR 25.72, exhibited comparable mass loss (*p* > 0.05 for all comparisons). TRIS samples with the largest SVR ratio exhibited significantly higher relative mass loss, compared to identical SVR, HCl and SBF samples (*p* = 0.01, 0.024, indicated by *), and when compared to TRIS samples in 16.5 and 9.06 SVR groups (*p* = 0.01, 0.02, indicated by †); (**B**) The mass loss is plotted against the SVR for each individual sample, in a scatter plot; (**C**) Linear regression analysis of all data points, with coefficient of determination (R^2^) for Tris, HCl and SBF groups; (**D**) Linear regression analysis of only the highest SVR samples from each group. Statistical analysis: average mass loss of each SVR group was compared to the average of the lowest SVR group, and each pH group (HCl, Tris, SBF) to the average of the lowest pH group (HCl). Comparisons were made with ANOVA, using Games–Howell post hoc analysis, with * and † indicating *p*-values below 0.05, and ** and †† indicating a *p*-value below 0.01.

**Figure 5 jfb-10-00054-f005:**
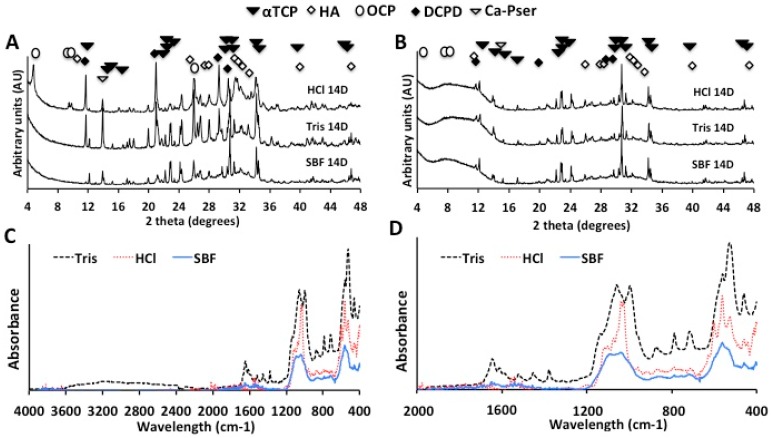
X-ray diffractogram, and infrared spectroscopy (FTIR-ATR) of PMC after soaking in HCl (0.02 M, pH 2.7), TRIS (1 M, pH 4.0), or SBF (pH 7.4). (**A**) The diffractograms of surface mineral reveal the presence of DCPD under acidic conditions (HCl, Tris), and in lesser amounts in SBF. OCP was found, only in HCl samples. Hydroxyapatite was also present in HCl samples; (**B**) Diffractograms from the entire sample (bulk) revealed that phase changes and reprecipitation occurred mostly at the surface, as the (crystalline) bulk material was primarily αTCP; (**C**,**D**) The FTIR spectra of surface mineral, scraped from the surface of the smallest SVR discs from each group, over the range of (**C**) 400–4000 cm^−1^, and (**D**) the region from 400 to 2000 cm^−1^ is expanded.

**Figure 6 jfb-10-00054-f006:**
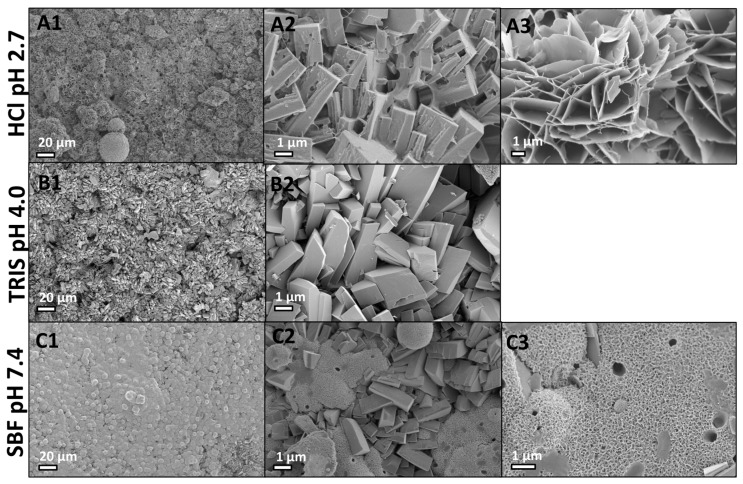
SEM images of PMC sample surfaces after 14 days in acidic (1 M HCl, or 0.2 M Tris), or neutral (SBF) conditions. (**A1**) HCl sample surfaces were covered with plate and cuboid shaped reprecipitants; (**B1**) TRIS sample surfaces were covered, exclusively, with cuboids; (**C1**) SBF samples were completely covered in spherical agglomerations, of small, plate-like morphology, with limited areas where cuboids protruded (like an exposed underlying layer); (Column 2; **A2**,**B2**,**C2**) Higher magnification image of cuboid reprecipitants, observed on HCl (**A2**), TRIS (**B2**), and SBF (**C2**) samples; (Column 3; **A3**,**C3**) Higher magnification image of plate-like reprecipitants, observed on HCl (**A3**) and SBF (**C3**) samples; plates were not observed on TRIS samples.

**Figure 7 jfb-10-00054-f007:**
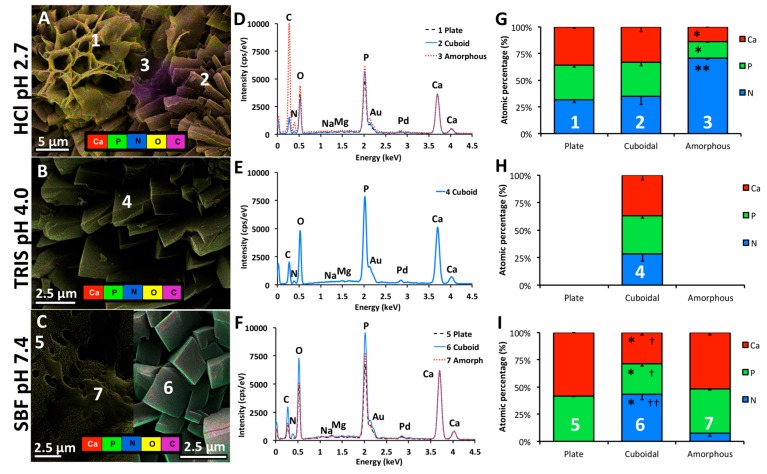
EDS analysis of surface morphology after dissolution testing. (**A**) HCl samples displayed three morphologies: inorganic plate-like (region 1), organic amorphous regions (region 3) similar to the amorphous surface of PMC immediately after setting and before starting the dissolution study, and cuboids (region 2); (**B**) TRIS samples exclusively displayed a mixed morphology that was difficult to distinguish (cuboid and/or DCPD) (region 4), which completely covered the surface; (**C**) SBF samples displayed two morphologies: spherical aggregates of small plate-like crystals (region 5), and cuboidal crystals (region 7). Small regions were observed that appeared amorphous (region 6), although the size and morphology differed from the amorphous regions found in HCl samples; (**D**–**F**) The spectra, obtained by EDS, of each corresponding region/ morphology (regions 1–7), with intensity scaled to the calcium peak; (**G**–**I**) Semi-quantitative comparison of elemental composition (spot analysis) at the regions indicated on images **A**–**C** (average of three sites). Statistical analysis compared the group mean, for each element of plate vs. cuboid and amorphous (indicated by *), and cuboid vs. amorphous (indicated by †), separately for each HCl, TRIS, and SBF. Comparisons were made by ANOVA, with Games–Howell post hoc analysis, with * and † indicating *p*-values below 0.05, and ** and †† indicating *p*-value below 0.01.

**Table 1 jfb-10-00054-t001:** PMC formulations used for compression testing or X-ray diffraction analysis.

Group	Phosphoserine (%)	αTCP (%)	Portland (%)	CS1 (%)	CS Day 28 (MPa)
Control *	0	100	0	0	-
1	31.0	79.0	0	0	49.2
2	31.0	67.8	1.2	0	43.5
3	0	98.3	1.7	0	10.4
4	31.0	67.8	0	1.2	38.6

* Control group was only included in X-ray diffraction studies.

**Table 2 jfb-10-00054-t002:** PMC sample dimensions, and surface area to volume ratio (SVR) for degradation testing.

Group	Measured SVR	Thickness (mm)	Diameter (mm)
2A	8.98	3	18
2B	16.67	2	6
2C	24.06	1	6

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
