# Peer review of "Phosphoserine Functionalized Cements Preserve Metastable Phases, and Reprecipitate Octacalcium Phosphate, Hydroxyapatite, Dicalcium Phosphate, and Amorphous Calcium Phosphate, during Degradation, In Vitro"

_jfb, 2019, doi:10.3390/jfb10040054_

Round 1

Reviewer 1 Report

This is review of the entitled manuscript “Phosphoserine functionalized cements preserve metastable phases, and reprecipitate octacalcium phosphate, hydroxyapatite, dicalcium phosphate, and amorphous calcium phosphate, during degradation, in vitro.” by Joseph Lazraq Bystrom and Michael Pujari-Palmer.

 The sample compressive strength is over 40 MPa and i think the mechanical strength was satisfied as bio-material. XRD patterns and IR spectra results were clear for sample phase transformation. Dissolution study and incubation is well selected as strong acid, weak acid and in vitro.

However, a little rewrite for clarity and flow by the authors is needed. In my opinion, it is not suitable publishable in this version, requires minor revisions as indicated.

Figure 6

Fig.6 caption is unclear.

I know about the picture information is described in the text, but, please add information in caption about which sample on D, E, F, G, H photo. One by one.

EDS analysis

Reference No. 69 discuss carbonate in the calcium phosphate samples. Carbonate in calcium phosphate is one of important point, therefore, please discuss about Carbon intensity in the sample based on EDS results more.

3 Please add Ref in Introduction section that is the investigation of interaction between phosphoserine and calcium phosphate.

Tanaka, H., Miyajima, K., Nakagaki, M., & Shimabayashi, S. (1991). Incongruent dissolution of hydroxyapatite in the presence of phosphoserine. Colloid and polymer science, 269(2), 161-165.

SHIMABAYASHI, S., & TANIZAWA, Y. (1990). Formation of hydroxyapatite in the presence of phosphorylated polyvinylalcohol as a simplified model compound for mineralization regulator phosphoproteins. Chemical and Pharmaceutical Bulletin, 38(7), 1810-1814.

Please describe your investigation originality point in the manuscript.

Author Response

Reviewer comment: This is review of the entitled manuscript “Phosphoserine functionalized cements preserve metastable phases, and reprecipitate octacalcium phosphate, hydroxyapatite, dicalcium phosphate, and amorphous calcium phosphate, during degradation, in vitro.” by Joseph Lazraq Bystrom and Michael Pujari-Palmer.

The sample compressive strength is over 40 MPa and i think the mechanical strength was satisfied as bio-material. XRD patterns and IR spectra results were clear for sample phase transformation. Dissolution study and incubation is well selected as strong acid, weak acid and in vitro.

***However, a little rewrite for clarity and flow by the authors is needed. In my opinion, it is not suitable publishable in this version, requires minor revisions as indicated.

AUTHOR RESPONSE: The authors thank the reviewer for this helpful comment. The text below has been modified as follows to improve clarity.

Reviewer comment: Fig.6 caption is unclear. I know about the picture information is described in the text, but, please add information in caption about which sample on D, E, F, G, H photo. One by one.

AUTHOR RESPONSE: The authors thank the reviewer for this helpful comment. The text has been modified as follows to improve clarity: (1) The figures have been relabeled into columns, column 1 is a low magnification overview of each sample; column 2 is higher magnification images of specifically the cuboid reprecipitants; and column 3 is higher magnification images of specifically the plate-like reprecipitants. (2) Specific description of each image is now provided, in short, concise text, when referring to each column (i.e. “Higher magnification image of cuboid reprecipitants, observed on HCl (A2), TRIS (B2), and SBF (C2) samples.”).

The modified caption for Figure 6 is shown below with revised text underlined or colored in red.
“SEM images of PMC sample surfaces after 14 days in acidic (1M HCl, or 0.2M Tris), or neutral (SBF) conditions. (A1) HCl sample surfaces were covered with plate and cuboid shaped reprecipitants. (B1) TRIS sample surfaces were covered, exclusively, with cuboids. (C1) SBF samples were completely covered in spherical agglomerations, of small, plate-like morphology, with limited areas where cuboids protruded (like an exposed underlying layer). (Column 2; A2, B2, C2) Higher magnification image of cuboid reprecipitants, observed on HCl (A2), TRIS (B2), and SBF (C2) samples. (Column 3; A3, C3) Higher magnification image of plate-like reprecipitants, observed on HCl (A3) and SBF (C3) samples; plates were not observed on TRIS samples.”

Reviewer comment: Reference No. 69 discuss carbonate in the calcium phosphate samples. Carbonate in calcium phosphate is one of important point, therefore, please discuss about Carbon intensity in the sample based on EDS results more.

AUTHOR RESPONSE: The authors acknowledge the reviewer’s point, however calcium carbonate is not present in any samples from the present study, and there is no evidence to suggest carbonate substitution in PMCs. While the authors draw attention to the interesting parallels between calcium carbonate and calcium phosphate systems in the introduction/ discussion, the authors believe that the carbon intensity information provided by EDS cannot be used to draw any further conclusions, for the following reasons: 1. Calcium carbonate is not present, as confirmed by XRD (crystalline calcite peaks is absent) and FTIR analysis (amorphous and crystalline carbonate peaks are absent); 2. Carbon is a lighter element, like oxygen, and may give inaccurate (semi-quantitative) results when measured by EDS, particularly for unpolished and uneven surfaces. If the reviewer can provide specific suggestions on what aspects of carbonate in calcium phosphate they recommend discussing (i.e. lattice substitution), the authors would be happy to include more information/discussion on this. Please advise.

Reference: Newbury*, D. E. and Ritchie, N. W. (2013), Is Scanning Electron Microscopy/Energy Dispersive X‐ray Spectrometry (SEM/EDS) Quantitative?. Scanning, 35: 141-168. doi:10.1002/sca.21041

Reviewer comment: Please add Ref in Introduction section that is the investigation of interaction between phosphoserine and calcium phosphate.

Tanaka, H., Miyajima, K., Nakagaki, M., & Shimabayashi, S. (1991). Incongruent dissolution of hydroxyapatite in the presence of phosphoserine. Colloid and polymer science, 269(2), 161-165.

SHIMABAYASHI, S., & TANIZAWA, Y. (1990). Formation of hydroxyapatite in the presence of phosphorylated polyvinylalcohol as a simplified model compound for mineralization regulator phosphoproteins. Chemical and Pharmaceutical Bulletin, 38(7), 1810-1814.

AUTHOR RESPONSE: The authors have added the suggested citations, and additional new citations, addressing how phosphoserine affects precipitation and dissolution of calcium salts. The following text has been added to the introduction, including the recommended citations (#59, and #55). “The amino acid phosphoserine plays a pivotal physiological role in calcium sequestration [46], mineralization and calcium phosphate precipitation [47], and tissue adhesion/cohesion [48]. Acidic non-collagenous proteins involved in mineralization in vertebrates, such as bone sailoprotein (BSP), osteopontin (OPN), and dentin phosphoprotein (DPP), are rich in phosphorylated and carboxyl- containing amino acids [47]. Paradoxically, phosphoserine can both accelerate, and arrest, precipitation/mineralization. In solution, phosphoserine chelates calcium and binds directly to crystal faces to interrupt crystal growth [37,47]. When phosphoserine is bound, embedded as matrix components (i.e. phosphatidyl-serine [46,49], or within proteins in saliva (i.e. statherin [50,51]) or dairy products (i.e. casein [52,53])), it can prevent mineralization in supersaturated fluids by chelating calcium, stabilizing amorphous calcium phosphate complexes, and direct binding to crystal surfaces. Alternatively, bound phosphoserine can facilitate mineralization by increasing the local concentration of sequestered ions, and ionic bonding [54,55]. Mutation studies have confirmed that phosphorylation is required for these activities, and acts by increasing the binding affinity an order of magnitude [51,54,56,57]. Phosphoserine also participates in the dissolution of calcium salts by directly etching the crystal surface, leading to substitution of surface inorganic phosphates with organic phosphate, and ionic chelation via the formation of calcium phosphoserine salt [56,58,59].“

Reviewer comment: Please describe your investigation originality point in the manuscript.

AUTHOR RESPONSE: The authors have added the following text to the discussion section:

The novelty of the present work includes the first report of: 1) PMC dissolution rate, in vitro; 2) surface remodeling and reprecipitation/mineralization in fluids that mimic specific physiological environments (i.e. sites of active resorption, or remineralization); and 3) how the chosen calcium salt affects the physiochemical properties of PMC (i.e. rate of transformation into hydroxyapatite and stabilization of metastable phases like αTCP) in liquids.

Reviewer 2 Report

General comments:

It should be a space between the last word of each sentence/phrase and the reference bracket []. Please fix this in the whole manuscript. The captions of the figures are not clear. The captions should not be a part of the main text (!). They should have different font sizes and should be placed as a caption under each related figure. For example Lines, 128-136 is supposed to be the caption of Figure 1, but now it looks like a part of the main text. Please fix this in the whole manuscript. There is no need to explain the plotting symbols of the figures in the captions! Each figure on itself should be presented clear enough and should be readable for the readers. The legends should be explained on the figures and not in the captions. Please add a suitable legend box with the required information for all figures if it is needed and remove the symbol explanations in the captions. The quality of all the figures is very low. Please replace them with high-quality versions. The curves in Figure1,2 and etc. should be plotted with colors in addition to different symbols. In this way, the plots will be readable for the readers who use the colored format of the manuscript. remove extra information in the figures captions. For instance in Figure 1-B we know that they are force-displacement curves, there is no need to mention this in the captions like: "Representative force/displacement curves". Please fix this in the whole manuscript. In Table 1, what does the group 0 means?! Can you better name the experimental groups?  Group-0 can be misleading to the readers. Lines 472 to 502 should be justified (same start and end points of lines).

Material and Methods:

A schematic image (or real one) of the test-setup with the dimensions of the specimen and the required information should be added. An image of one representative sample before and after the compression test should be added. What was the frequency of the test machine for collecting the data? This information also should be added to the paper.

Author Response

Reviewer comment: It should be a space between the last word of each sentence/phrase and the reference bracket []. Please fix this in the whole manuscript.

AUTHOR RESPONSE: This has been done. We thank the reviewer for catching this error.

Reviewer comment: The captions of the figures are not clear. The captions should not be a part of the main text (!). They should have different font sizes and should be placed as a caption under each related figure. For example Lines, 128-136 is supposed to be the caption of Figure 1, but now it looks like a part of the main text. Please fix this in the whole manuscript. There is no need to explain the plotting symbols of the figures in the captions! Each figure on itself should be presented clear enough and should be readable for the readers. The legends should be explained on the figures and not in the captions. Please add a suitable legend box with the required information for all figures if it is needed and remove the symbol explanations in the captions.

AUTHOR RESPONSE: The caption text size and formatting has been corrected, the authors apologize for this oversight. The plotting symbol information has been removed from all figures.

Reviewer comment: The quality of all the figures is very low. Please replace them with high-quality versions.

AUTHOR RESPONSE: The authors are attaching high quality images (300 DPI), but cannot include these in the word/PDF file, since the size is too large to upload to the MDPI website. Therefore, the images in the word document will not be high quality. Please advise on whether the new image files (.tiff) are acceptable.

Reviewer comment: The curves in Figure1,2 and etc. should be plotted with colors in addition to different symbols. In this way, the plots will be readable for the readers who use the colored format of the manuscript. remove extra information in the figures captions. For instance in Figure 1-B we know that they are force-displacement curves, there is no need to mention this in the captions like: "Representative force/displacement curves".

AUTHOR RESPONSE: The lines in Figure 1, and 4 have been colorized as suggested. The authors feel that colorizing the XRD diffractograms in Figures 2 and 5 would make the figures unclear, rather than clearer. The information in the XRD data is rather dense, and adding colors may overwhelm or confuse the reader. In Figure 1 the text “Representative force/displacement curves” has been changed to “Representative force/displacement curves showing comparable modulus (slope), and brittle failure for groups 1, 2, and 4.” The phrase could not be deleted, since we must provide some explanation for figure 1B, so we have clarified the statement and made it less extraneous. Please advise on whether the reviewer suggests additional changes.

Reviewer comment: Please fix this in the whole manuscript. In Table 1, what does the group 0 means?! Can you better name the experimental groups?  Group-0 can be misleading to the readers.

AUTHOR RESPONSE: The authors thank the reviewer for this helpful comment. Group 0 has been changed to “group Control”. Since the “group Control” was only included in the XRD study and no other parts of this work, the authors feel it would be confusing for the reader if it was included in the table as if it were a regular group “i.e. group 1”. Therefore, the authors have changed this group name to “group Control” to indicate that this group differs from the other groups, in that it was only included as a control for XRD analysis.

Reviewer comment: Lines 472 to 502 should be justified (same start and end points of lines).

AUTHOR RESPONSE: The authors thank the reviewer for this helpful comment. The text has been justified and we have reviewed the entire document to ensure it is properly formatted.

Reviewer comment: Material and Methods: A schematic image (or real one) of the test-setup with the dimensions of the specimen and the required information should be added. An image of one representative sample before and after the compression test should be added. What was the frequency of the test machine for collecting the data? This information also should be added to the paper. 

AUTHOR RESPONSE: The authors agree, and have made a new figure (graphical abstract) as suggested.
